# Effects of a Dietary Multi-Strain Probiotic and Vaccination with a Live Anticoccidial Vaccine on Growth Performance and Haematological, Biochemical and Redox Status Indicators of Broiler Chickens

**DOI:** 10.3390/ani12243489

**Published:** 2022-12-10

**Authors:** Anna Arczewska-Włosek, Sylwester Świątkiewicz, Katarzyna Ognik, Damian Józefiak

**Affiliations:** 1Department of Animal Nutrition and Feed Science, National Research Institute of Animal Production, 32-083 Balice, Poland; 2Department of Biochemistry and Toxicology, Faculty of Animal Sciences and Bioeconomy, University of Life Sciences in Lublin, 20-950 Lublin, Poland; 3Department of Animal Nutrition, Faculty of Veterinary Medicine and Animal Science, Poznań University of Life Sciences, 60-637 Poznań, Poland

**Keywords:** anticoccidial vaccine, probiotic, broiler chickens, growth performance, blood variables

## Abstract

**Simple Summary:**

Coccidiosis, a parasitic disease caused by protozoa of the genus *Eimeria*, is one of the most frequently investigated enteric poultry diseases, primarily due to its ubiquity and severe negative effects on the economic efficiency of the poultry industry. Immunoprophylaxis with live anticoccidial vaccines is regarded as an effective tool to control this parasitic disease; however, there is a great reluctance to use this approach in broilers, primarily because of reports of transient reduced performance due to the state of ‘mild coccidial infection’ associated with anticoccidial vaccines. In this context, the administration of some feed additives may be useful as supplementation to improve the health status and immune response. Therefore, the present study aimed to evaluate the effect of dietary supplementation with a probiotic on growth performance, oocyst shedding, and selected blood parameters in broilers vaccinated with a live oocyst vaccine.

**Abstract:**

A total of 256 male Ross 308 chickens were assigned to four treatments in a 2 × 2 factorial design with two levels of the anticoccidial vaccine (ACV) Livacox T (none or 1 × dose) with or without dietary supplementation with the probiotic Protexin^®^ (P). The growth performance parameters for the test periods (1–21, 22–42, and 1–42 d) and oocyst per gram (OPG) at weekly intervals were analysed. Blood samples were collected at 16 post-vaccination (pv) days to measure selected haematological, biochemical, redox, and immunological parameters. ACV administration worsened the performance parameters of the chickens for 1–21 d pv, while supplementation with P reduced this negative effect with a significant improvement in 1–21 d body weight gain and feed conversion ratio. ACV administration increased % phagocytic cells (%PC), phagocytic index (PI), respiratory burst activity, proportion of monocytes, and activities of aspartate aminotransferase (AST) and lactate dehydrogenase, while it decreased the catalase activity and concentration of malondialdehyde and peroxides. The dietary administration of P significantly increased counts of red blood cells and white blood cells and increased %PC and PI, while it decreased the heterophil proportion, heterophil/lymphocyte ratio (*p* = 0.059), and alanine aminotransferase and AST activities. The oocyst counts were comparable in all sampling periods, except on 14 d pv, as supplementation with P significantly decreased 14 d OPG, thus indicating a positive influence of P on immunity development. In conclusion, dietary supplementation with P led to improved performance, better immunity, and benefits in health status in broilers vaccinated with the ACV, without interfering with the circulating vaccine strains.

## 1. Introduction

Coccidiosis, caused by the intestinal protozoan parasite *Eimeria* species, has long been recognised as one of the most common and economically significant parasitic diseases in chickens [1,2,3,4]. The multiplication and growth of the coccidian parasite in the intestinal epithelium during infection cause diverse symptoms such as diarrhoea, haemorrhage, or even death in the severe form of coccidiosis or malabsorption, reduced performance, and enteritis in the subclinical form [1,5,6]. All of these effects compromise the welfare of birds and reduce the economic effectiveness of poultry production. According to a recent report by Blake et al. [1], the global cost of coccidiosis in chickens is estimated at approximately GBP 10.4 billion at 2016 prices, equivalent to GBP 0.16/chicken produced. Therefore, several efforts are being made globally to develop effective strategies to prevent or mitigate the negative effects of coccidiosis. Presently, the primary methods for controlling coccidiosis in chickens include the routine use of coccidiostats or anticoccidial vaccines (ACVs) based on live *Eimeria* strains [3]. However, both of these methods have limitations. For coccidiostats, the emergence of resistance of *Eimeria* strains to anticoccidials and the growing demand of customers for chemoprophylaxis-free poultry production have prompted the need to identify, develop, and implement other strategies to control coccidiosis.

Immunoprophylaxis of broilers with live ACVs, even though regarded as an effective approach to control coccidiosis and commonly used in the production of layers and breeders, still raises concerns of transient performance deterioration that may not be compensated during the relatively short lifespan of chickens [7,8,9]. ACVs function by inoculating a small number of oocysts of the specified species, which undergo their life cycle in the chicken gut. The second or third subsequent infection with the circulating oocysts should result in the development of acquired immunity to *Eimeria* species included in the vaccine, as the immunity is species-specific [10]. This process is sometimes accompanied by a state similar to mild subclinical coccidiosis, as the intestinal integrity may be compromised following replication of the vaccine oocysts in the intestinal epithelium; this results in a diminished absorptive intestinal surface area, malabsorption, and inflammation, and it can also be a predisposing factor for bacterial secondary enteritis [6,8].

In this context, nutritional methods as a supplementation approach to maintaining a healthy intestinal tract with balanced microflora might mitigate the abovementioned side effects of ACVs [11,12]. Probiotics, also known as direct-fed microbials, consist of non-pathogenic bacteria or fungi that beneficially act on animal health and growth performance through different modes of action such as stimulation of the immune system, modulation of the gut microbial ecosystem through the production of primary and secondary metabolites with antibacterial properties, and competitive exclusion of pathogens; these effects support the development of beneficial microflora, together with having a positive impact on the structural modulation of the intestinal epithelium and maintenance of the intestinal barrier integrity as well as increasing digestive enzyme activity and improving digestion [13,14,15]. Probiotics used in poultry are composed of one or multi-strain microbial species, which mainly belong to the genera *Lactobacillus*, *Streptococcus*, *Bacillus*, *Enterococcus*, *Pediococcus*, *Aspergillus*, and *Saccharomyces* [15,16]. The effectiveness of single [17,18,19,20] or multi-strain probiotics [21] was confirmed in broilers challenged with *Eimeria* and in birds vaccinated against coccidiosis [16,22]. However, most studies on probiotics and ACVs included a subsequent challenge with *Eimeria* to evaluate the combined effects of a probiotic and vaccine as a protective measure against coccidiosis. There is less research on the impact of probiotics in vaccinated birds that are not exposed to coccidiosis outbreaks or are experimentally challenged, although this situation is frequently encountered in poultry production conditions.

Protexin^®^ (Probiotics International Ltd., Lopen Head, Somerset, UK) is a multi-strain probiotic preparation containing seven bacterial and two yeast strains. This bacterial and fungal combination preparation has been proven to enhance the immune response to Newcastle disease virus, lower the counts of caecal *Escherichia coli* [23], improve growth performance, and lower the levels of proinflammatory cytokines in broilers challenged with multiple bacterial species including avian pathogenic *E. coli*, *Salmonella enteritidis*, and *Salmonella typhimurium* [24]. However, this specific preparation has not been evaluated in chickens exposed to *Eimeria* oocysts.

Based on previous findings, we hypothesized that dietary supplementation with the probiotic Protexin^®^ may reduce the potential negative side effects of ACVs on the growth performance of broiler chickens, without impairment of the circulating vaccine strain, which is crucial for the development of acquired immunity against coccidiosis. Thus, the present study aimed to evaluate the effects of Protexin^®^ on broiler chickens vaccinated against coccidiosis in terms of their growth performance; profile of oocyst output of vaccine origin; and health status reflected by haematological, biochemical, redox, and immunological variables.

## 2. Materials and Methods

### 2.1. Birds, Diets, and Experimental Design

The animal experimental procedures were approved by the Second Local Ethical Committee on Animal Testing, Cracow, Poland. The study was designed as a 2 × 2 factorial experiment with 8 replicate pens of 8 male Ross 308 chickens per treatment. Treatments included a lack or a single dose of ACV (Livacox T^®^; Biopharm Co., Prague, Czech Republic, administered at 1 d of age) with or without dietary supplementation with the multi-strain probiotic Protexin^®^ (Probiotics International Ltd., Lopen Head, Somerset, UK).

A total of 256 one-day-old male Ross 308 broiler chickens, obtained from a commercial hatchery (Daniela Kożuch Poultry Hatchery, Łężkowice, Poland), were randomly assigned to treatments at 1 d of age. The experimental period lasted for 42 days. The birds were reared under standard environmental conditions, with the temperature maintained from 32 °C at 1 d of age to 21 °C at 21 d of age.The birds were placed in floor pens with a total floor space of 0.76 m^2^ and equipped with 2 nipple-cup drinkers and a trough feeder. A bedding of wood shavings was used to ensure the recirculation of oocysts of vaccine origin. To avoid the transition of oocysts and reduce the risk of contamination of the unvaccinated groups, polyvinyl chloride sheets were used as barriers between the pens.

The birds were fed with maize–soybean meal basal diets in a mash form, and the diets were formulated to meet or exceed the nutritional requirements of broiler chickens for the starter (1–21 d of age) or grower-finisher (22–42 d of age) feeding phase (Table 1) [25]. The birds were provided water and fed *ad libitum*. The diets were free of antibiotic growth promoters and coccidiostats. Based on the chemical composition of raw feed materials, the content of nutrients in the basal diets was calculated, and the value of metabolizable energy was estimated according to equations from European Tables [26].

### 2.2. Experimental Factors

At 1 d of age, 50% of the birds were administered a single, recommended dose of live, attenuated ACV (Livacox^®^ T; Biopharm Research Institute of Biopharmacy and Veterinary Drugs, Prague, Czech Republic) before the chicks were placed in the designated pens. The single vaccine dose (0.01 mL), which contained 300–500 sporulated oocysts each of *Eimeria acervulina*, *Eimeria maxima*, and *Eimeria tenella*, was suspended in 0.24 mL distilled water and administered per os. Birds from the unvaccinated groups received an identical volume of distilled water.

In the treatment groups assigned to receive the probiotic bacteria, Protexin^®^ was mixed with basal diets at the dose of 0.15 or 0.10 g/kg of feed in the starter or grower-finisher feeding phase, respectively. The composition of Protexin^®^ was as follows: *Lactobacillus plantarum*, 1.89 × 10^10^ cfu/kg; *Lactobacillus delbrueckii* ssp. *Bulgaricus*, 3.09 × 10^10^ cfu/kg; *Lactobacillus acidophilus*, 3.09 × 10^10^ cfu/kg; *Lactobacillus rhamnosus*, 3.09 × 10^10^ cfu/kg; *Bifidobacterium bifidum*, 3.00 × 10^10^ cfu/kg; *Streptococcus salivarius* ssp. *Thermophilus*, 6.15 × 10^10^ cfu/kg; *Enterococcus faecium*, 8.85 × 10^10^ cfu/kg; *Aspergillus oryzae*, 7.98 × 10^9^ cfu/kg; and *Candida pintolopesii*, 7.98 × 10^9^ cfu/kg.

### 2.3. Sample Collection and Analytical Procedure

The standard AOAC methods [27] were used to analyse the basal diets for moisture (method 930.15), crude protein (method 984.13), crude fat (method 920.39), ash (method 942.05), amino acids (method 982.30), calcium (method 968.08), and total phosphorus content (method 965.17).

The feed intake was recorded weekly, and the birds were weighed at the age of 1, 21, and 42 d. The body weight gain (BWG), feed intake (FI), and feed conversion ratio (FCR) were calculated for the starter, grower-finisher, and entire experimental period (1–42 d). FCR was calculated as kg feed/kg BWG, and the data were corrected for mortality. All growth performance data were analysed on a pen basis (*n* = 8).

To demonstrate the profile of oocyst shedding, the number of oocysts per gram of excreta (OPG) was determined using a concentration McMaster technique with a McMaster counting chamber [28]. OPG was calculated using pooled faecal samples collected from each replicate pen (*n* = 8) at 7, 14, 21, 28, and 35 d post-vaccination (pv). To obtain the pooled faecal sample, several fresh faecal samples from different locations in the pen were collected and homogenized for the analysis of OPG from each pen (replicate) separately. The OPG values were logarithmically transformed [log_10_ (OPG + 1)] to create a normal distribution.

At 16 d of age, the blood samples from 6 chickens per experimental group (1 chicken/replicate; *n* = 6) were collected from the wing vein into heparinized tubes and tubes without the anti-coagulant. The blood samples were then centrifuged at 3000× *g* for 10 min to obtain plasma and serum.

The following haematological parameters were analysed: haematocrit (Ht) and haemoglobin (Hb) levels, and red blood cell (RBC) and total white blood cell (WBC) count with leukograms [29]. The heterophil-to-lymphocyte ratio (H/L) was calculated as a parameter for the stress response [30,31].

Immunological analyses included the determination of phagocytic activity of leukocytes against the strain *Staphylococcus aureus* 209P, represented as the proportion of phagocytic cells (% PC) and as the phagocytic index (PI) [32]. The test of nitroblue tetrazolium (NBT) reduction to formazan was used to assess the respiratory burst activity of heterophils [33]. The turbidimetric method [32] was used to determine the lysozyme concentration.

The following biochemical indices were determined in the blood serum samples by using commercial kits developed by Cormay Co. (PZ Cormay Inc., Lomianki, Poland): total protein (TP), total cholesterol (TC), triacylglycerols (TG), and glucose (GLU) concentrations as well as enzyme activities: aspartate aminotransferase (AST), alanine aminotransferase (ALT), lactate dehydrogenase (LDH), and alkaline phosphatase (ALP).

As reported previously [34], the following indicators of redox status were determined in chicken blood plasma samples: concentrations of lipid peroxides (LOOH); malondialdehyde (MDA); ferric reducing ability of plasma (FRAP), as a parameter of the antioxidant potential; and activities of superoxide dismutase (SOD) and catalase (CAT).

### 2.4. Statistical Analysis

The data were analysed using STATISTICA version 13.3 (StatSoft Inc., Tulsa, OK, USA) software. Two-way analysis of variance (ANOVA) was used to determine the main effects of treatments such as ACV and dietary supplementation with the probiotic as the main factors and their interactions. One-way ANOVA was used to analyse the effect of probiotic supplementation on OPG results in vaccinated chickens. Duncan’s multiple range post hoc test was used to determine the differences between the treatments, and the effects were considered significant at a probability level of *p* ≤ 0.05.

## 3. Results

Performance results obtained in the starter, grower-finisher, and entire rearing period are presented in Table 2. The effects of ACV, probiotic supplementation, and their interaction on growth performance were significant only in the starter period. Regarding the independent effects, ACV significantly lowered FI, reduced BWG, and led to a higher FCR for the 1–21 d period, while dietary supplementation with probiotics significantly increased BWG for 1–21 d regardless of the vaccination status. The interaction of ACV × P was significant for 1–21 d FCR. Although P supplementation had no effect on FCR in unvaccinated birds, it improved FCR in the vaccinated ones to the level obtained in the unvaccinated groups.

Figure 1 presents the profile of oocyst shedding throughout the experimental period in the vaccinated groups of birds. The largest increase in OPG for the unsupplemented group was observed on 14 d pv, and the OPG value gradually decreased in the subsequent sampling periods. The highest value of OPG in the supplemented group was found on 7 d pv. On 14 d pv, the OPG value was significantly lower in the supplemented group than in the unsupplemented group. For all other sampling periods, the OPG was not affected by P supplementation (*p* > 0.05).

Table 3 presents the results for haematological parameters analysed from blood samples collected at 16 d of age. The unvaccinated birds receiving P showed a significant increase in RBC counts, while in vaccinated birds, the effect of P supplementation had no significant effect on this parameter (ACV × P; *p* ≤ 0.05). Regardless of ACV administration, P supplementation significantly increased the WBC count, decreased the proportion of heterophils in the leukogram, and decreased the H/L ratio (*p* = 0.059). The birds of both vaccinated groups showed an increase in the monocyte proportion in the leukogram (*p* ≤ 0.05). Haematocrit and haemoglobin levels and the proportions of lymphocytes, basophils, and eosinophils in the leukogram were not significantly influenced by the studied factors.

Significant effects of ACV administration were observed on the increased values of % PC, PI, and NBT in the vaccinated birds. Similarly, P supplementation increased the % PC. The unvaccinated birds receiving P supplementation in the diet showed a higher PI, while in the vaccinated ones, P supplementation did not further increase the PI values (Table 4).

Table 5 shows the effects of experimental factors on the biochemical indices. ACV × P interactions were significant for AST activity and TP and GLU concentrations. Although vaccination resulted in a significantly higher AST activity and TP concentration in the unsupplemented group, P supplementation in vaccinated birds decreased the AST and TP values to levels comparable to those observed in both unvaccinated groups in whom P supplementation did not affect these parameters. In the vaccinated group, P supplementation increased the serum GLU concentration, while the feed additive did not cause a significant difference in unvaccinated birds. Regarding independent effects, ACV administration increased the LDH activity and serum TG concentration, but it lowered the ALP activity and CHOL concentration. P supplementation also had significant effects on the ALT activity and CHOL concentration, as in both vaccinated and unvaccinated birds, P supplementation decreased the values of these parameters.

Regarding the redox status parameters (Table 6), ACV administration significantly decreased the CAT activity, reduced the blood LOOH concentration, and increased the FRAP values (*p* = 0.054). P supplementation showed a significant effect on the CAT activity, as both vaccinated and unvaccinated chickens receiving P supplementation showed lower CAT activity. The ACV × P interaction affected the MDA concentration. P supplementation decreased the MDA concentration in the unvaccinated birds but led to a significantly higher MDA concentration in the vaccinated ones. None of the experimental factors significantly affected the SOD activity.

## 4. Discussion

Similar to the results of our previous study [35], the performance data obtained for 1–21 d pv in the present experiment, i.e., reduced FI and BWG and increased FCR, support the possible deterioration of growth performance in this period due to ACV administration. However, no significant differences regarding growth performance were observed at the end of the experiment, thus indicating that the vaccinated birds underwent compensatory growth following the initial decrease in growth performance after vaccination. Dietary supplementation with P resulted in improved BWG and FCR for the 1–21 d period, thus alleviating the negative side effects of ACV administration. Ritzi et al. [16] reported comparable effects of ACV administration and probiotic supplementation, as birds vaccinated with Immucox I (CEVA Animal Health Inc., Guelph, Ontario, Canada) and supplemented with a probiotic product (PoultryStar, BIOMIN GmbH, Herzogenburg, Austria) containing *Enterococcus*, *Bifidobacterium*, *Pediococcus*, and *Lactobacillus* species showed significantly greater weight gain as compared to vaccinated and non-supplemented ones; this result suggests that the addition of probiotics helped the birds to counter growth deterioration associated with ACV administration.

In the current study, we expected that dietary supplementation with the probiotic preparation would not exert a coccidiostatic effect, which could interfere with the circulation of oocysts of the vaccine strains and impair the development of acquired immunity. The observed OPG values in the group of vaccinated chickens supplemented with P indicate that this feed additive lacks coccidiostatic or coccidiocidal mode of action; this makes it suitable for use in birds vaccinated with live oocysts. According to the description given by the vaccine manufacturer, immunity should be acquired within 14 d pv. In the present study, this was reflected in the oocyst shedding profile that steadily declined from 14 d pv in the unsupplemented vaccinated group. In the P-supplemented group, the highest OPG value was observed at 7 d pv and then continuously decreased with significantly lower OPG at 14 d pv when compared with the unsupplemented group. This finding indicates that P supplementation can accelerate the development of immunity. The improved production of *Eimeria*-specific antibodies following P supplementation could be the mechanism underlying this effect. This hypothesis can be confirmed based on the results of the study of Lee et al. [36], where the supplementation of a probiotic consisting of *Pediococcus acidilactici* and *Saccharomyces boulardii* in birds infected with *E. acervulina* and *E. tenella* increased the level of serum *Eimeria*-specific antibodies along with decreased oocyst shedding. In the current study, the enhanced humoral immunity probably reduced inflammation and intestinal tissue damage by replicating a lower number of oocysts of the vaccine strains, which could result in improved functionality of the intestinal absorptive surface area and eventually in improved growth performance. Moreover, ACV administration may affect the intestinal microflora, which could also be linked to decreased growth performance. Orso et al. [9] demonstrated that, in ceca, ACV administration lowered the percentage of *Bacteroidetes* phylum that comprises genera producing short-chain fatty acids known for improving intestinal health and increased the percentage of deleterious *Proteobacteria* phylum. Probiotic supplementation could help maintain homeostasis of the intestinal microflora. In chickens challenged with *Eimeria*, the probiotic product of the whole cell wall of the yeast *Pichia guilliermondii* ameliorated the coccidial infection-induced caecal microflora shift by decreasing the population of *Salmonella* and *E. coli* and preventing a decrease in *Lactobacillus* population [37]. Wang et al. [38] performed 20 × overdosed ACV administration to simulate coccidial infection, and dietary supplementation with *Bacillus subtilis* increased bacterial species number in non-vaccinated broilers but decreased bacterial species in vaccinated broilers. *B. subtilis* positively altered the microbial profile by promoting *Rikenella microfusus* abundance, which decreased the proportion of other bacteria in the *Firmicutes* phylum and caused a lower abundance of *Faecalibacterium*, *Blautia*, *Alkaliphilus*, *Ruminococcus*, and *Clostridium*.

Heterophils and lymphocytes are the most abundant WBC types in birds, and they play an essential role in innate and adaptative immunity, respectively. Heterophils are highly phagocytic and are involved in the acute inflammatory response as the first line of immune defence, mainly during the first hours following an immunological challenge [39,40]. In contrast, lymphocytes are involved in humoral adaptive immunity (B cells) and cell-mediated adaptive immunity (T cells). An elevated H/L ratio is generally regarded as an indicator of physiological stress in birds. Moreover, birds with a lower H/L ratio show better survival rates or resistance to *Salmonella* infection [41,42]. In this context, the haematological parameters observed at 16 d pv confirmed the beneficial effects of P supplementation, as it caused a significant increase in RBC and WBC counts and lowered the proportion of heterophils in the leukogram. Furthermore, in groups supplemented with P, the H/L ratio showed a tendency to decrease (*p* = 0.059), and the lymphocyte proportion increased. This profile of haematological variables obtained in the present study suggests that the probiotic participates in the development of better adaptive immune response and a reduction in stress. Similarly, Mohsin et al. [4] reported that the supplementation of *Lactobacillus plantarum* to broilers receiving a subunit vaccine containing *E. maxima* immune mapped protein-1 and subsequently challenged with coccidial infection resulted in increased WBC and RBC counts.

In the current study, ACV administration significantly elevated the values of parameters related to the phagocytic activity of leukocytes, such as % PC, PI, and NBT, at 16 d pv; this finding indicates an increase in the innate immunity response to the vaccine’s strains of oocysts. P supplementation significantly increased the % PC in both vaccinated and unvaccinated birds, but it increased PI only in the unvaccinated birds. Stringfellow et al. [22] reported that the effect of the probiotic PoultryStar and vaccine Coccivac B combination varied depending on the age of the birds and was the most pronounced at 14 d pv. At this period, ACV induced a significantly higher oxidative burst of heterophils, while the probiotic added to the water for vaccinated broilers caused a significantly higher oxidative burst of monocytes and heterophils and stimulated a cell-mediated immune response, as reflected in significantly higher lymphocyte proliferation. In the present study, the observed reduction in oocyst shedding at 14 d pv in birds receiving the probiotic may be linked to a higher immune response.

Coccidiosis often triggers an immune response that results in the generation of reactive free radicals such as nitric oxide (NO) and reactive oxygen species (ROS) [43,44]. An insufficient antioxidative defence system may negatively affect chickens and lead to intestinal necrosis because of oxidative damage. In broilers infected with *E. acervulina* [45] and *E. tenella* [31,32], oxidative stress was indicated as a factor because the increased NO and/or MDA concentration reduced SOD and increased CAT activities. As vaccination is regarded as a challenge similar to the course of mild coccidiosis, we expected that the corresponding profile of parameters would reflect the redox status. ACV administration significantly decreased the CAT activity, reduced the levels of serum peroxides, and increased the FRAP values (*p* = 0.054), with no effect on SOD activity. These results do not indicate the occurrence of oxidative stress during blood sampling. Moreover, P supplementation decreased the CAT activity and modulated the MDA concentration differently (by increasing and decreasing the MDA concentration in the vaccinated and unvaccinated groups, respectively); this result is difficult to explain. The hypothetical explanation for these unexpected results may be the fact that constant exposure to low doses of attenuated oocysts, along with the increase in the values of parameters related to the phagocytic activity, mainly in the respiratory burst activity of heterophils (NBT), may lead to a different adaptive mechanism that relies on CAT utilization to neutralize peroxides.

The activities of enzymes such as AST, ALT, LDH, or ALP are commonly used to verify whether the tested nutritional factors interfere negatively with the liver function [46]. In the present study, P supplementation significantly lowered the activity of AST and ALT in vaccinated birds, indicating the hepatoprotective properties of the probiotic. Similarly, supplementation with the probiotic *L. plantarum* (1 × 10^8^ CFU) significantly decreased AST, ALT, and LDH activities in broilers infected with *E. tenella* [19].

P supplementation lowered the TC concentration and increased the GLU concentration in the serum of vaccinated birds. The positive cholesterolemic effect of Protexin^®^ was reported previously [47,48], both in chickens challenged with *Salmonella enteritidis* [49] and in cockerels [23]. Among the proposed mechanisms, the one responsible for the hypocholesterolemic effect of probiotics may be the ability of certain lactic acid bacteria to produce the bile salt hydrolase enzyme that deconjugates bile salts. As deconjugated bile salts are less soluble at low pH, they are not absorbed efficiently in the intestine, leading to greater faecal excretion [23,50].

To accurately identify the mechanisms underlying the positive effects of the tested probiotics in vaccinated birds, further experiments are required to investigate their effects on gut histology, gut barrier integrity, the gut microbiota profile, and additional parameters of immunity.

## 5. Conclusions

The concept of using probiotics as a supportive tool for immunoprophylaxis with live ACV is a promising approach, as shown by a recent study. Supplementation with a multi-strain probiotic mitigates the negative effects of ACV on the growth performance and indicates the stimulating effects of the probiotic on the immune response, as reflected in parameters related to phagocytic activity or OPG. Probiotic supplementation also provided health benefits by increasing RBC and WBC counts and reducing the concentration of TC or liver enzymatic activity. The probiotic used in the present study did not affect the circulation of the vaccine’s oocysts, which is crucial for immune development; thus, this probiotic is recommended to decrease the potential adverse effects associated with ACV administration.

## Figures and Tables

**Figure 1 animals-12-03489-f001:**
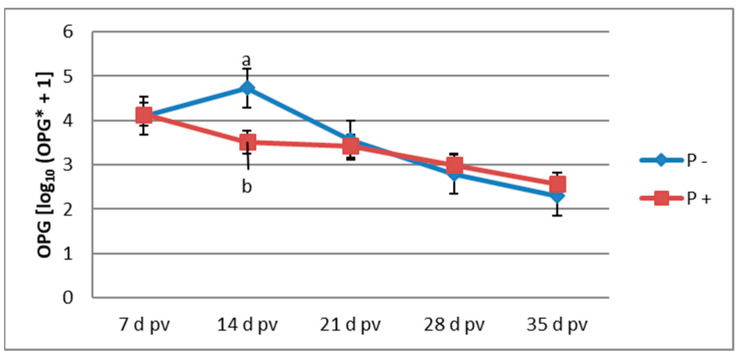
Oocyst shedding in the vaccinated groups [log_10_ (OPG + 1]; (*n* = 6); a, b—mean values followed by different letters differ significantly at *p* ≤ 0.05; OPG*—oocysts per gram; pv—post-vaccination; P—probiotic preparation.

**Table 1 animals-12-03489-t001:** Composition of experimental diets for starter (1–21 d) and grower-finisher (22–42 d) feeding phases and nutrient content in basal feed mixtures.

Ingredient [g/kg]:	Starter	Grower-Finisher
Maize	579.3	597.5
Soybean meal	360	323
Soybean oil	18	38
Limestone	16	16
Monocalcium phosphate	14.5	14.0
Sodium chloride	3	3
DL-Methionine	2	2
L-Lysine hydrochloride	1.2	1.5
Vitamin-mineral premix *	6	5
Calculated nutritional value per kg of feed:		
Metabolizable energy (MJ/kg)	12.3	13.0
Analysed chemical composition (g/kg):		
Dry matter	891	892
Crude ash	70.2	58.1
Crude protein	221	212
Crude fat	19.7	32.5
Crude fibre	24.5	22.0
Calcium	10.6	9.57
Phosphorus	7.82	7.25
Asp	22.09	20.34
Tre	8.54	7.66
Ser	11.56	10.19
Glu	39.37	36.97
Pro	12.53	11.81
Gly	9.13	8.33
Ala	10.74	10.19
Val	9.96	9.13
Ile	8.89	8.09
Leu	18.39	17.26
Tyr	8	6.61
Fen	11.23	10.13
His	5.41	4.91
Lys	12.99	11.59
Arg	15.82	12.68
Cys	3.26	3.15
Met	5.39	5.04
Trp	2.18	1.93

* Each kilogram of the vitamin-mineral premix contained the following: vitamin A—2,000,000 IU; vitamin D3—500,000 IU; vitamin E—7000 IU; vitamin K3—600 mg; vitamin B1—400 mg; vitamin B2—1400 mg; vitamin B6—1000 mg; vitamin B12—8 mg; Ca-pantothenate—2000 mg; niacin—8000 mg; folic acid—200 mg; biotin—16 mg; choline chloride—29,480 mg; manganese—16,000 mg; zinc—12,000 mg; iron—12,000 mg; copper—3000 mg; iodine—400 mg; selenium—50 mg.

**Table 2 animals-12-03489-t002:** Effects of the experimental factors on the performance in the starter (1–21 d) and grower-finisher (21–42 d) feeding phases and the entire experimental period (1–42 d).

		1–21 d of Age			22–42 d of Age			1–42 d of Age		
Factors		BWG	FI	FCR	BWG	FI	FCR	BWG	FI	FCR
**ACV**	**P**	(g)	(g)	(g/g BWG)	(g)	(g)	(g/g BWG)	(g)	(g)	(g/g BWG)
−	−	636	1009	1.59 b	1723	3423	1.99	2305	4482	1.95
	+	664	1042	1.57 b	1741	3453	1.98	2341	4565	1.95
+	−	568	926	1.63 a	1763	3518	1.99	2289	4489	1.96
	+	625	987	1.58 b	1752	3455	1.97	2324	4496	1.94
SEM		9.52	14.1	0.005	21.3	44.1	0.012	23.7	51.4	0.012
**Significance (** ** *p* ** **-value)**										
Effects	ACV	0.001	0.011	0.000	0.568	0.602	0.962	0.742	0.771	0.983
	P	0.009	0.073	0.000	0.934	0.862	0.591	0.481	0.678	0.686
Interaction	ACV × P	0.344	0.585	0.013	0.738	0.619	0.757	0.986	0.724	0.569

a, b—mean values within the same column followed by different letters differ significantly at *p* ≤ 0.05; ACV—anticoccidial vaccine; P—probiotic preparation; BWG—body weight gain; FI—feed intake; FCR—feed conversion ratio; SEM—standard error of mean.

**Table 3 animals-12-03489-t003:** Haematological parameters and leukogram in the chicken blood collected at 16 d of age.

Factors		RBC	WBC	Ht	Hb	H	L	MONO	EOS	BASO	H/L
**VAC**	**P**	(10^12^ L^−1^)	(10^9^ L^−1^)	(%)	(g^−1^)	(%)	(%)	(%)	(%)	(%)	
−	−	1.56 b	16.3	31.8	6.68	31.7	65.2	1.00	1.50	0.67	0.51
	+	1.93 a	18.6	29.4	5.82	23.5	70.7	1.00	3.50	1.33	0.34
+	−	1.76 a	15.8	29.1	6.37	26.8	65.3	2.33	3.67	1.83	0.43
	+	1.83 a	18.7	28.4	6.15	21.2	69.5	3.33	3.50	2.50	0.33
SEM		0.042	0.309	0.572	0.179	1.738	1.827	0.366	0.716	0.340	0.035
**Significance (** ** *p* ** **-value)**											
Effects	VAC	0.460	0.523	0.099	0.976	0.288	0.896	0.011	0.473	0.094	0.535
	P	0.004	0.000	0.170	0.147	0.048	0.214	0.451	0.543	0.328	0.059
Interactions	VAC × P	0.034	0.287	0.429	0.372	0.708	0.861	0.451	0.473	1.000	0.585

a, b—mean values within the same column followed by different letters differ significantly at *p* ≤ 0.05; VAC—anticoccidial vaccine; P—probiotic bacteria; SEM—standard error of mean; RBC—red blood cells; WBC—white blood cells; Ht—haematocrit; Hb—haemoglobin; H—heterophils; L—lymphocytes; MONO—monocytes; EOS—eosinophils; BASO—basophils; H/L—heterophil/lymphocyte ratio.

**Table 4 animals-12-03489-t004:** Immunological indices of chicken blood collected at 16 d of age.

Factors		LYSOZYME	% PC	PI	NBT: Positive Heterophils
**VAC**	**P**	(mg L^−1^)	(10^9^ L^−1^)		(%)
−	−	1.33	41.3	5.09 c	24.9
	+	1.31	43.8	5.75 b	25.3
+	−	1.34	46.2	6.32 a	40.1
	+	1.29	46.5	6.44 a	37.2
SEM		0.037	0.512	0.123	1.93
**Significance (** ** *p* ** **-value)**					
Effects	VAC	0.947	0.000	0.000	0.000
	P	0.649	0.023	0.001	0.653
Interactions	VAC × P	0.909	0.063	0.018	0.569

a–c—mean values within the same column followed by different letters differ significantly at *p* ≤ 0.05; VAC—anticoccidial vaccine; P—probiotic bacteria; SEM—standard error of mean; % PC—percentage of phagocytic cells; PI—phagocytic index; NBT—nitroblue tetrazolium reduction test.

**Table 5 animals-12-03489-t005:** Biochemical indices of chicken blood collected at 16 d of age.

Factors		AST	ALT	LDH	ALP	TP	TG	TC	GLU
**VAC**	**P**	(U/ L)	(U/ L)	(U/ L)	(U/ L)	(g/L)	(mmol/L)	(mmol/L)	(mmol/L)
−	−	232 b	3.53	1330	6.03	29.5 b	1.78	4.29	9.33 ab
	+	242 b	2.57	1353	3.93	28.5 b	1.68	3.50	8.78 b
+	−	280 a	2.75	1610	3.55	38.2 a	3.51	3.31	8.64 b
	+	243 b	1.90	1761	3.42	27.6 b	2.79	3.13	9.75 a
SEM		5.66	0.233	61.7	0.399	1.032	0.322	0.120	0.156
**Significance (** ** *p* ** **-value)**									
Effects	VAC	0.014	0.105	0.004	0.048	0.003	0.028	0.001	0.610
	P	0.143	0.046	0.420	0.133	0.000	0.506	0.008	0.310
Interactions	VAC × P	0.018	0.893	0.551	0.183	0.000	0.615	0.073	0.006

a, b—mean values within the same column followed by different letters differ significantly at *p* ≤ 0.05; VAC—anticoccidial vaccine; P—probiotic bacteria; SEM—standard error of mean; AST—aspartate aminotransferase; ALT—alanine aminotransferase; LDH—lactate dehydrogenase; ALP—alkaline phosphatase; TP—total protein; TG—triacylglycerol; TC—total cholesterol; GLU—glucose.

**Table 6 animals-12-03489-t006:** Redox status indices of chicken blood collected at 16 d of age.

Factors		FRAP	SOD	CAT	LOOH	MDA
**VAC**	**P**	(μmol/L)	(U/mL)	(U/mL)	(μmol/L)	(μmol/L)
−	−	100	28.4	3.21	2.71	0.454 a
	+	95.4	29.0	2.74	2.57	0.367 b
+	−	110	28.7	2.77	1.61	0.331 b
	+	122	28.3	2.37	1.62	0.487 a
SEM		4.69	0.260	0.089	0.140	0.018
**Significance (*p*-value)**						
Effects	VAC	0.054	0.670	0.007	0.000	0.966
	P	0.670	0.898	0.004	0.716	0.197
Interactions	VAC × P	0.352	0.357	0.801	0.703	0.000

a, b—mean values within the same column followed by different letters differ significantly at *p* ≤ 0.05; VAC—anticoccidial vaccine; P—probiotic bacteria; SEM—standard error of mean; FRAP—total antioxidant potential, as a ferric reducing ability of plasma; SOD—superoxide dismutase; CAT—catalase; LOOH—lipid peroxides; MDA—malondialdehyde.

## Data Availability

The data presented in this study are available within the article.

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
