# Peer review of "Effects of a Dietary Multi-Strain Probiotic and Vaccination with a Live Anticoccidial Vaccine on Growth Performance and Haematological, Biochemical and Redox Status Indicators of Broiler Chickens"

_animals, 2022, doi:10.3390/ani12243489_

Round 1

Reviewer 1 Report

CommentsThe objective of the manuscript was to provide a comprehensive explanation of the use of probiotics in chicken farms to combat coccidiosis in the feed and their effects on mRNA gene expression as growth promotor, immunomodulatory, and serum chemistry of chickens.

Manuscript is attractive however; below mentioned points should be improved as well as Native English person needs to revise the manuscript

Line 16-18: In this context, the administration of some of the feed additives may be useful as health status and immune supporting factor “modify to” In this context, the administration of some feed additives may be useful as health status and immune supporting factor

Line 28-29: ACV resulted in an increase   in % phagocytic cells (%PC), phagocytic index (PI), the respiratory burst activity “modify to” ACV increased in % phagocytic cells (%PC), phagocytic index (PI), the respiratory burst activity

Line 29-31: The dietary administration of P had a significant effect on increased counts for red blood cells “modify to” The dietary administration of P significantly increased counts for red blood cells

Line 34-35: as supplementation with P significantly decrease 14 d OPG “modify to” as supplementation with P significantly decreased 14 d OPG

Line 42-44: Coccidiosis, caused by Eimeria species of intestinal parasites, has long been recognised as one of the most common and economically significant parasitic diseases in chickens “here” add most recent references such as   https://doi.org/10.1016/j.ygeno.2021.10.019 and http://dx.doi.org/10.29261/pakvetj/2020.043:

Line: during infection includes as diverse symptoms as diarrhoea, haemorrhage or even death in the course of severe coccidiosis, or malabsorption, reduced performance, and enteritis in the subclinical form “here” add most recent reference such as DOI: 10.29261/pakvetj/2021.009

Line 52-54: Rephrase the sentence: Nowadays, the primary method of control of coccidiosis  in chickens is the use of coccidiostats, but also anticoccidial vaccines (ACV) based on live Eimeria strains

Line 63: Thereafter “modify to” after that

Line 73-80: Probiotics, also known as direct-fed microbials, consist of non-pathogenic bacteria or fungi, that beneficially act on animal health and growth performance by different          modes of action such as stimulation of the immune system, modulation of the gut microbial ecosystem through the production of primary and secondary metabolites with anti-bacterial properties, and competitive exclusion of pathogens which, in turn, supports the development of beneficial microflora, along with having a positive impact on the structural modulation of the intestinal epithelium, maintenance of the intestinal barrier integrity, increasing digestive enzyme activity, and improving digestion “add” the most recent references

Line 81-82: Probiotics used  in poultry are composed of one or multi-strain microbial species, belonging to the genera Lactobacillus, Streptococcus, Bacillus, Enterococcus, Pediococcus, Aspergillus and Saccharomyces “add” most recent references

Line 92-187: What’s about the concentration of oocysts dose got/bird? And for each test how many birds or replicate used? Explain and improved it clearly

Line 162-164: The  heterophil to lymphocyte ratio (H/L) was calculated as a parameter relating to a stress  reaction “modify to” The heterophils to lymphocyte ratio (H/L) was calculated as a parameter relating to a stress reaction

Line 195-197: Rephrase the sentence: While the supplementation with P had no effect ondid not affect FCR in unvaccinated birds, in vaccinated ones resulted in  improved FCR to the level obtained in the unvaccinated groups.

Line 252-253: probiotic administration resulted in decrease values of these parameters “changed to” probiotic administration resulted in decreased values of these parameters

Line 347: The activities of enzymes such as AST, ALT, LDH or ALP “add “the comma after LDH

Line 348: The activities of enzymes such as AST, ALT, LDH, or ALP are commonly used to    verify if tested nutritional factors did not interfere negatively with liver function “here” add the recent references

Line 364-366: Rephrase the sentence: The administration of probiotics resulted also in benefits in health by its influence on an increase in RBC, WBC counts, and decrease in TC concentration or enzymatic activity.

Line 368: What’s about the future research may be conduct following your study?

Line 390-480: Check the references carefully following journal format reference style

Comment: How probiotics is useful against poultry coccidiosis? What’s about the possible mechanism of probiotics alone or in combination with vaccine? Directly or indirectly effect against Eimeria parasite?

Comment: How can you compare this study to others (novelty) as many studies have been completed on probiotics (single or multi strains) alone and in combination with vaccine/others anticoccidial against Eimeria parasite?

Author Response

REVIEWER 1- responses

Authors thank the Reviewer for the thorough reading of the manuscript and very helpful remarks and proposed corrections that helped us to improve our manuscript.

We have addressed all comments and suggestions raised by the Reviewer, and have modified the paper accordingly (changes highlighted in yellow). Also, after the revision, the manuscript underwent thorough language correction by the native speaker via the professional proofreading service.

Comments: The objective of the manuscript was to provide a comprehensive explanation of the use of probiotics in chicken farms to combat coccidiosis in the feed and their effects on mRNA gene expression as growth promotor, immunomodulatory, and serum chemistry of chickens.

Manuscript is attractive however; below mentioned points should be improved as well as Native English person needs to revise the manuscript

Line 16-18: In this context, the administration of some of the feed additives may be useful as health status and immune supporting factor “modify to” In this context, the administration of some feed additives may be useful as health status and immune supporting factor

Response: The text was modified.

Line 28-29: ACV resulted in an increase   in % phagocytic cells (%PC), phagocytic index (PI), the respiratory burst activity “modify to” ACV increased in % phagocytic cells (%PC), phagocytic index (PI), the respiratory burst activity

Response: The text was modified.

Line 29-31: The dietary administration of P had a significant effect on increased counts for red blood cells “modify to” The dietary administration of P significantly increased counts for red blood cells

Response: The text was modified.

Line 34-35: as supplementation with P significantly decrease 14 d OPG “modify to” as supplementation with P significantly decreased 14 d OPG

Response: The text was modified.

Line 42-44: Coccidiosis, caused by Eimeria species of intestinal parasites, has long been recognised as one of the most common and economically significant parasitic diseases in chickens “here” add most recent references such as   https://doi.org/10.1016/j.ygeno.2021.10.019 and http://dx.doi.org/10.29261/pakvetj/2020.043:

Response: The suggested references were added.

Line: during infection includes as diverse symptoms as diarrhoea, haemorrhage or even death in the course of severe coccidiosis, or malabsorption, reduced performance, and enteritis in the subclinical form “here” add most recent reference such as DOI: 10.29261/pakvetj/2021.009

Response: The suggested reference was added.

Line 52-54: Rephrase the sentence: Nowadays, the primary method of control of coccidiosis  in chickens is the use of coccidiostats, but also anticoccidial vaccines (ACV) based on live Eimeria strains

Response: The sentence was rephrased to: “Nowadays, the primary methods of control of coccidiosis in chickens areis the routine use of coccidiostats, but also or anticoccidial vaccines (ACV) based on live Eimeria strains”.

Line 63: Thereafter “modify to” after that

Response: The text was modified.

Line 73-80: Probiotics, also known as direct-fed microbials, consist of non-pathogenic bacteria or fungi, that beneficially act on animal health and growth performance by different          modes of action such as stimulation of the immune system, modulation of the gut microbial ecosystem through the production of primary and secondary metabolites with anti-bacterial properties, and competitive exclusion of pathogens which, in turn, supports the development of beneficial microflora, along with having a positive impact on the structural modulation of the intestinal epithelium, maintenance of the intestinal barrier integrity, increasing digestive enzyme activity, and improving digestion “add” the most recent references  

Response: The recent references were added.

Line 81-82: Probiotics used  in poultry are composed of one or multi-strain microbial species, belonging to the genera LactobacillusStreptococcus, Bacillus, Enterococcus, Pediococcus, Aspergillus and Saccharomyces “add” most recent references

Response: The recent references were added.

Line 92-187: What’s about the concentration of oocysts dose got/bird? And for each test how many birds or replicate used? Explain and improved it clearly

Response: Each bird got an one dose of the vaccine, as the producer recommended. One dose of vaccine contains coccidian oocysts of:

Eimeria acervulina: 300 - 500

Eimeria tenella: 300 - 500

Eimeria maxima: 300 - 500

 The growth performance data and OPG were analyzed on a pen basis (n=8), while for blood analysis 6 chickens per experimental group (1 chicken/ 1 replicate; n=6) were chosen. The appropriate information was added to the manuscript.

Line 162-164: The  heterophil to lymphocyte ratio (H/L) was calculated as a parameter relating to a stress  reaction “modify to” The heterophils to lymphocyte ratio (H/L) was calculated as a parameter relating to a stress reaction

Response: Authors decided to remain the orginal form “heterophil to lymphocyte ratio”, as it is commonly used in the scientific papers, such as:

Gross WB, Siegel HS. Evaluation of the heterophil/lymphocyte ratio as a measure of stress in chickens. Avian Dis. 1983 Oct-Dec;27(4):972-9.

Cotter PF. An examination of the utility of heterophil-lymphocyte ratios in assessing stress of caged hens. Poult Sci. 2015 Mar;94(3):512-7. doi: 10.3382/ps/peu009.

Thiam M, Wang Q, Barreto Sánchez AL, Zhang J, Ding J, Wang H, Zhang Q, Zhang N, Wang J, Li Q, Wen J, Zhao G. Heterophil/Lymphocyte Ratio Level Modulates Salmonella Resistance, Cecal Microbiota Composition and Functional Capacity in Infected Chicken. Front Immunol. 2022 Apr 14;13:816689. doi: 10.3389/fimmu.2022.816689.

Line 195-197: Rephrase the sentence: While the supplementation with P had no effect ondid not affect FCR in unvaccinated birds, in vaccinated ones resulted in  improved FCR to the level obtained in the unvaccinated groups.

Response: The sentence were rephrased.

Line 252-253: probiotic administration resulted in decrease values of these parameters “changed to” probiotic administration resulted in decreased values of these parameters

Response: The text was modified.

Line 347: The activities of enzymes such as AST, ALT, LDH or ALP “add “the comma after LDH

Response: The text was modified.

Line 348: The activities of enzymes such as AST, ALT, LDH, or ALP are commonly used to    verify if tested nutritional factors did not interfere negatively with liver function “here” add the recent references

Response: The recent reference was added.

Line 364-366: Rephrase the sentence: The administration of probiotics resulted also in benefits in health by its influence on an increase in RBC, WBC counts, and decrease in TC concentration or enzymatic activity.

Response: The sentence were rephrased.

Line 368: What’s about the future research may be conduct following your study?

Response: The appropriate suggestions for future research in the field are placed in L 402-405.

Line 390-480: Check the references carefully following journal format reference style

Response: We checked the reference style and prepare it with the help of citation management software.

Comment: How probiotics is useful against poultry coccidiosis? What’s about the possible mechanism of probiotics alone or in combination with vaccine? Directly or indirectly effect against Eimeria parasite?

Response: The relevant information was added to the Discussion section (L 314-338).

Comment: How can you compare this study to others (novelty) as many studies have been completed on probiotics (single or multi strains) alone and in combination with vaccine/others anticoccidial against Eimeria parasite?

Response: The relevant part was added to the Introduction section (L 86-100).

Reviewer 2 Report

Reviewer #1: In this manuscript, the researchers tried to explain about the The effectiveness of multi-strain probiotic in the diet of broiler chickens vaccinated against coccidiosis. It is interesting work and can be accepted after revision.

-     The grammar errors should be checked in the whole manuscript.

-       In abstract, the first four lines should be summarized.

-       In introduction, the main objective has been repeated so it should be refined.

-       Some recent and relevant articles may be added as thousands of articles have been published on this topic.

-       Conclusion should be refined as it is not properly written as per results.

Author Response

REVIEWER 2- responses

Authors are grateful for all valuable comments and improvements concerning our manuscript, given by the Reviewer. We addressed all comments and concerns raised by the Reviewer to significantly improve the quality and the clarity of the manuscript, as follows:

In this manuscript, the researchers tried to explain about the The effectiveness of multi-strain probiotic in the diet of broiler chickens vaccinated against coccidiosis. It is interesting work and can be accepted after revision.

-     The grammar errors should be checked in the whole manuscript.

Response: After the revision, the manuscript underwent thorough language correction by the native speaker via the professional proofreading service.

-       In abstract, the first four lines should be summarized.

Response: This was corrected according to the suggestion.

-       In introduction, the main objective has been repeated so it should be refined.

Response: In this part, we consider both the hypothesis and the research objective.

-       Some recent and relevant articles may be added as thousands of articles have been published on this topic.

Response: The recent references were added (highlighted).

-       Conclusion should be refined as it is not properly written as per results.

Response: The Conclusion section was modified.

Reviewer 3 Report

The authors studied the topic of The Effectiveness of Multi-Strain Probiotic in the Diet of Broiler Chickens Vaccinated against Coccidiosis. The article is well written, clearly stated and reads smoothly. But I feel that the research ideas are not novel, in other word, not innovative enough. As mentioned in the background, Introduction, coccidiosis vaccine reduces broiler production performance, and the addition of probiotics to feed improves broiler growth performance. Both of them have been extensively studied. In this study, the authors used a 2*2 experimental design, mainly to observe whether there was an interaction between the two factors. Judging from the results, the interaction between the two factors is not significant enough. From the perspective of the mechanism of action, the main reason may be that the two factors have no basis for interaction. If the probiotic factor is replaced by improved nutrient levels, the interaction may be better. Therefore, authors should focus on why probiotics are used to mitigate the negative effects of coccidiosis vaccines in the introduction section. In terms of the interaction analysis, the results should be presented more pertinently, in response to the title and research theme. The primary role of probiotics is to improve gut health and microbial composition, but these indicators were not tested in this study.

What is the IF in the Abstract.

The classification of material methods is not clear, such as chemical analysis methods should not be placed in 2.1.

The amino acid content units in Table 1 are wrong

The specific P value should be given in the Tables. Change the Unit l to L

Table 5... If there is an interaction, it is not appropriate to use two pairs analysis.

Author Response

REVIEWER 3- responses

Authors are very grateful for all the valuable comments and improvements concerning our manuscript, given by the Reviewer. We addressed all comments and concerns raised by the Reviewers to improve the quality and clarity of the manuscript significantly. Also, the manuscript underwent thorough language correction by the native speaker via the professional proofreading service.

The authors studied the topic of The Effectiveness of Multi-Strain Probiotic in the Diet of Broiler Chickens Vaccinated against Coccidiosis. The article is well written, clearly stated and reads smoothly. But I feel that the research ideas are not novel, in other word, not innovative enough. As mentioned in the background, Introduction, coccidiosis vaccine reduces broiler production performance, and the addition of probiotics to feed improves broiler growth performance. Both of them have been extensively studied. In this study, the authors used a 2*2 experimental design, mainly to observe whether there was an interaction between the two factors. Judging from the results, the interaction between the two factors is not significant enough. From the perspective of the mechanism of action, the main reason may be that the two factors have no basis for interaction. If the probiotic factor is replaced by improved nutrient levels, the interaction may be better. Therefore, authors should focus on why probiotics are used to mitigate the negative effects of coccidiosis vaccines in the introduction section. In terms of the interaction analysis, the results should be presented more pertinently, in response to the title and research theme. The primary role of probiotics is to improve gut health and microbial composition, but these indicators were not tested in this study.

Response: Thank you for the comments. The benefits that may provide probiotics in the nutrition of vaccinated birds are given in the Introduction section, and some discussion on the probiotic mode of action that may underlie the obtained results was added. Also, we changed the title to reflect the manuscript's content better. Moreover, previously we investigated the effect of crude protein level and/ or herbal extract blend (Arczewska-WÅ‚osek, A.; ÅšwiÄ…tkiewicz, S.; Kowal, J.; Józefiak, D.; DÅ‚ugosz, J. The Effect of Increased Crude Protein Level and/or Dietary Supplementation with Herbal Extract Blend on the Performance of Chickens Vaccinated against Coccidiosis. Animal Feed Science and Technology 2017, 229, 65–72, doi:10.1016/j.anifeedsci.2017.04.021.). In the present study we focus only on the effect of probiotic.

What is the IF in the Abstract.

Response: The text was corrected, as it should be PI (phagocytic index).

The classification of material methods is not clear, such as chemical analysis methods should not be placed in 2.1.

Response: The part of text on the chemical analysis methods of diets was moved to 2.3 section (Sample collection and analytical procedure).

The amino acid content units in Table 1 are wrong

Response: The units were corrected.

The specific P value should be given in the Tables. Change the Unit l to L

Response: The tables were corrected according to the suggestions.

Table 5... If there is an interaction, it is not appropriate to use two pairs analysis.

Response: Thank you for the comment. We tried to focus on the interaction effect.

Round 2

Reviewer 1 Report

Comment: Recheck the references including Journal names used in references carefully to maintain uniformaty following journal format

Author Response

Thank you for the comment. The reference list was corrected according to the journal requirements.

Authors thank the Reviewer for the thorough reading of the manuscript and very helpful remarks and proposed corrections that helped us to improve our manuscript. We are also very grateful for such quick review.

Reviewer 3 Report

The authors responded my concerns well.

Author Response

Response: Authors are very grateful to the Reviewer for such quick reviews and all valuable and constructive suggestions that helped us to improve our manuscript.